# Evaluation of Thermal Comfort Performance of a Vertical Garden on a Glazed Façade and Its Effect on Building and Urban Scale, Case Study: An Office Building in Barcelona

**Faezeh Bagheri Moghaddam \***, **Josep Maria Fort Mir**, **Isidro Navarro Delgado** and **Ernesto Redondo Dominguez**

Escola Tècnica Superior d'Arquitectura de Barcelona, Universitat Politècnica de Catalunya, 08028 Barcelona, Spain; Josep.Maria.Fort@upc.edu (J.M.F.M.); Isidro.Navarro@upc.edu (I.N.D.); Ernesto.Redondo@upc.edu (E.R.D.)

**\*** Correspondence: Faezeh.bagheri.moghaddam@upc.edu

**Abstract:** The aim of this paper is to investigate the thermal performance of vertical gardens by comparing the thermal comfort of bare (glazed) and green façades in the Mediterranean climate. The proposal consists of applying a vegetation layer on a glazed façade that could control solar radiation and reduce indoor air temperatures. This study investigates the thermal performance of green façades of an office building in the Mediterranean climate. For this purpose, the Gas Natural Fenosa Office Building as a case study was simulated, that is located on a site next to the coastline in Barcelona. Dynamic building energy simulation was used to determine and assess indoor thermal conditions and, for this reason, the IES VE as a simulation tool has been utilized. Thermal comfort was assessed through the adaptive comfort approach and results were analyzed and presented in the terms of indoor comfort conditions during occupied hours. As a result, the article shows that applying a green façade as a vegetation layer caused a reduction in the internal and external façade surface temperatures, as well as the indoor air temperature of the workplace. Additionally, enhancing indoor comfort in summer is closely associated with reducing the external surface temperature. In winter, it also protects the exterior surface from the low temperature of the outside, and all of this greatly increases thermal comfort performance.

**Keywords:** green façade; thermal comfort; air temperature; urban scale; building simulation; sustainability



## 1. Introduction

Green façades as vertical greening systems have many ecological and environmental benefits in the urban scale, and some of them can be highlighted in urban rehabilitation: improving air quality, reducing the urban heat island (UHI) impact, improving stormwater management, and absorbing air pollutants from the atmosphere [1–5]. Urban greenery, such as green façades, has become a significant issue in recent years because the majority of the world's population lives in cities [6], must deal with global carbon emissions rising by 70%, and accounts for nearly 70% of energy consumption. Moreover, there is a growing trend in both carbon emissions and energy consumption [7], and land conversion to urban areas is expected to triple by 2030 [8]. Vertical greenery systems allow for increased vegetation in urban contexts while taking up no street space, enhancing biodiversity, and indirectly improving urban appearance. Some examples of vertical greenery systems, which present some typologies of hanging greenery as a solution for improving the environmental sustainability of buildings, have been proposed [9].

Environmental issues have many significant impacts, including human health, citizens' quality of life, and urban economic efficiency [8]. Green façades can be important for building energy efficiency and also urban microclimate mitigation [10–12]. A microclimate

is described as any region where the climate differs from the surrounding region, and a large urban microclimate can affect not only temperatures but also rainfall, snowfall, wind, and air pressure. For all studies, six parameters of green façades have been suggested that should be considered in order to obtain an adequate quantification of thermal performance and microclimatic benefit. These parameters are solar radiation, air temperature, and wind speed in front of or away from the green façade and/or between the green façade and the wall [13].

Regarding previous studies, which revealed that the vegetation layer on the façade can reduce the temperature of external surfaces of the building envelope during summer [14,15], consequently, the green layer could improve indoor comfort in terms of air and surface temperature reduction [16]. Many parameters must be considered when investigating green façades, such as the orientation of the façades [17,18], as well as the water distribution [15,19] and climate conditions. Researchers have demonstrated, through a sensitivity analysis, that solar radiation, wind speed, relative humidity, and outside air temperature are important for weather parameters [20]. Green façades are especially useful for high solar exposure walls and where ground-level space is restricted, or where aerial obstructions restrict tree growth [21]. Through investigating vertical greenery systems in terms of direct and indirect environmental resources, it is possible that green façade systems could reach a condition of comprehensive sustainability in a 25-year lifetime [22]. Comprehensive sustainability is linked to full-cycle sustainability, which indicates that the environmental impact of a product (or service) is thoroughly analyzed at every stage of its life. The concept of comprehensive sustainability must be taken into account as a holistic, topologic, and synergetic approach that does not allow for the dualism between people and nature, which is at the origin of our current difficulties.

Green façades, as a design feature in a warming environment, are able to cool internal building temperatures, reduce building energy usage, and encourage urban adaptation [13]. Furthermore, several studies have shown that vertical greenery systems have a positive effect on the building envelope in terms of thermal comfort, especially during cooling periods [23]. Some research about green façades in the Mediterranean region has shown a possible reduction in surface temperature of more than 10.8 °C [14]. In addition, researchers compared a green façade (with climbing plants) to a bare façade, finding that the surface temperatures of the green façade were up to 15.5 °C lower than those of the bare façades, while those of the interior walls were up to 1.7 °C lower [24]. It is possible that accurate characterization of the green façades will be required to better understand the contribution of such systems to the enhancement of hydrothermal conditions, the infrared radiation emitted and intercepted by the green canopy, as well as the relative humidity. Moreover, when it comes to vertical greenery systems, the surface temperature and the inside temperature of the substrate may provide valuable information about the usefulness and thermal efficiency [16].

The capacity of vegetation layers to cool is linked to the shading and evapotranspiration effect of plants [25]. Cooling is accomplished by the leaves on the façade, absorbing solar radiation (as a result of phototropism [26]) and shielding the back wall. Moreover, during the summer, a vegetation façade lowers the temperature by evaporating water from the foliage's surface [1].

Several factors influence the cooling efficiency of vertical greenery systems [27], including façade orientation, which is particularly important in green façades because of the evapotranspiration and the shadow created by plants [18,28]. High-density foliage coverage, creating a stagnant air layer (cavity) behind the foliage [29], using supporting system materials and their insulation effect, and plant species characteristics [30] can all be used to improve the insulation properties of vertical greenery systems.

With vertical greenery systems, the potential energy saving for air conditioning in Mediterranean areas can be up to 40–60% [25,31–35]. According to research, green façades can save 1.30, 0.84, and 0.71 kW/h of energy per day for an 8.22 m$^2$ flat on sunny, cloudy, and rainy days in summer, respectively, and can save up to 16% of the electricity consump-

tion for air conditioning in July, August, and September, which are the hottest months of the year [36].

Many investigations have been conducted to determine the performance of green façades and their effect on energy consumption, thermal efficiency, and temperature variation, which revealed that vertical greenery systems can reduce the energy demand for air conditioning by reducing indoor temperatures [25]. Further research is required to investigate building climate control by the potential contribution of green façades [37].

Glazed surfaces may have a major effect on occupants' thermal comfort for two reasons: their transparency enables solar radiation to enter the space and increase the glazed façade's inner surface temperature, and the temperature might differ greatly from temperatures of other surfaces, inducing long-wave radiant heat exchange and convective heat transfer to the adjacent space [38].

The aim of this study was to assess the thermal comfort efficiency of green façades in the Mediterranean climate by simulating and evaluating a pilot project developed in Barcelona using the IES VE software as a simulation tool. As the reduction in indoor temperature caused by a green layer is heavily affected by the building envelope layers, it is possible to infer that insulation content moderates the prevailing temperature differential between the inside and outside [35,39]. This study demonstrates how to use a green façade's cooling ability to improve thermal comfort while lowering energy consumption.

In Mediterranean regions, research on the exterior and interior surface temperatures of façades, and also the indoor air temperature of vertical greenery, revealed that this façade technology affects the microclimate, building thermal comfort, and energy use during the summer and winter [40–42]. While the number of green areas and other low-albedo surfaces must be maximized to reduce the UHI effect in cities [43], reducing air and radiant temperature by greenery systems at the urban scale directly affects outdoor thermal comfort as well [44].

## 2. Methodology

A green façade scenario was applied to the Gas Natural Fenosa Building, which is an office building, as a case study. This simulation was conducted by covering approximately half of the building façade with a 16 cm thick plant (ivy) layer, which could raise the R-value and reduce the U-value of the façade, as well as a 50 cm cavity between the glazed façade and the plant layer. Plants and their substrate on the façade would also increase the R-value, resulting in lower energy costs [45]. The dynamic Integrated Environmental Systems software (IES VE) was used to predict the thermal efficiency of the green façade in Barcelona. The IES subroutines, which are RadianceIES and Apache, assess the effect of a green façade on daylighting and thermal comfort. Thermal comfort is influenced by a number of factors. Environmental and personal factors are two types of those factors. Air temperature, humidity, radiant temperature, and air velocity are the environmental factors, and activity level and clothing are personal factors. According to the environmental factors, the results were analyzed to demonstrate the thermal comfort performance of the green façade in comparison with the bare façade (glazing façade).

According to the Spanish Regulations for Thermal Facilities in Buildings (RITE) (*Reglamento de Instalaciones Térmicas en los Edificios*), and the indoor air quality (IAQ) categories (IDA) that are classified based on building use, the office building is in IDA 2. With regard to this classification, the indoor air quality (IAQ) of office buildings must be good [46]. The RITE sets standards for thermal comfort in offices and the principles that were approved by the Occupational Risk Prevention Act (Law 21/1995 PRRLL). At the same time, the INSHT provides guidance for safe working practices in offices [47,48].

### 2.1. Climate Characterization

Barcelona is a Mediterranean coastal city with a Mediterranean climate. The Azores dominate the weather throughout the summer. Summers are hot and dry, with tempera-

tures averaging about 28 °C. Furthermore, the months of July and August are the hottest in the year.

When the Azores prime passes southwards in the winter, westerly winds with a little more rain prevail in the Mediterranean. Barcelona may be shielded from the wintry winds that often blow from the Pyrenees through Catalonia by the nearby mountains. In addition, temperatures in Barcelona seldom fall below 0 °C in the winter, and the average winter daytime temperature is around 13 °C.

Figures 1 and 2 depict the sun's direction, angle, and number of hours of sunlight in Barcelona, Spain. For green façade implementation, accurate knowledge of sun paths throughout the year and climatic conditions is needed, as well as knowledge of orientation, landscaping, summer shading, solar collector area, and the cost-effective use of solar trackers.

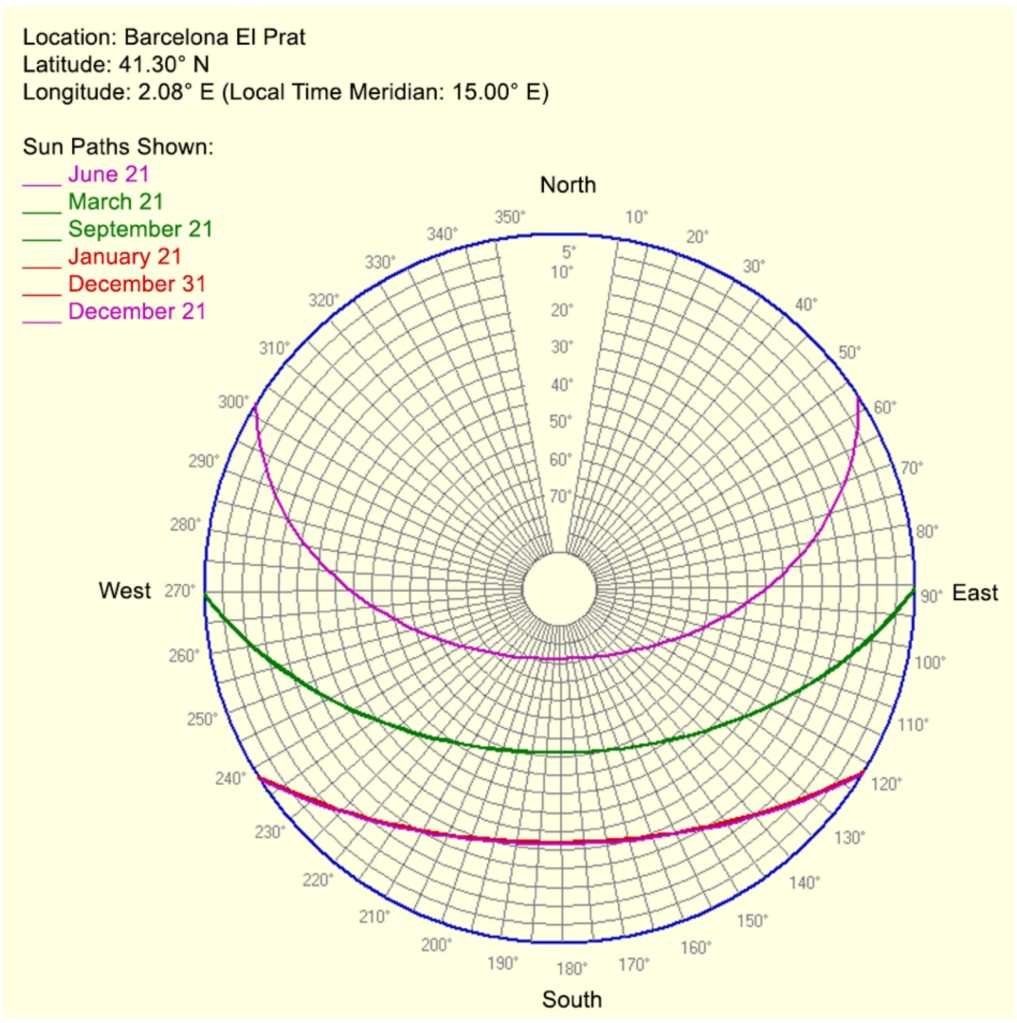

**Figure 1.** Barcelona sun path graph. © IES VE.

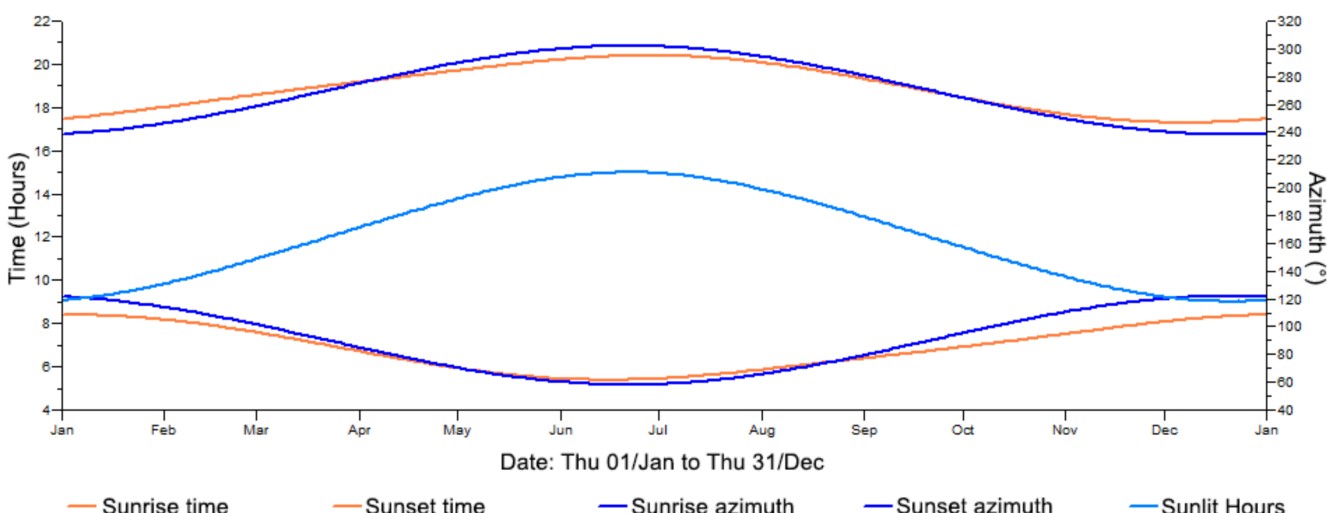

**Figure 2.** Graph of sun up/down parameters. © IES VE.

*2.2. Model Validation*

By comparing predicted and measured indoor space temperatures, the simulation results were used to validate the wall model (glazed and green wall). Knowing the variations in surface and indoor air temperature between the bare and green façades allowed for the estimation of other façade properties, such as a decrease in heat transfer through the façade (assuming a constant R-value for the façade itself) and thus the concurrent effective R-value of the vegetation layer. To adapt the direct solar radiation data to a vertical surface, the IES VE simulation technique was used to model and apply correlations between the two façades to the measured data. The most extreme days in the results, June 21 and December 21, were used for validation.

June 21 was chosen because the Northern Hemisphere has the longest duration of daylight and the sun takes the longest journey across the sky at the summer solstice, so there is more solar energy on this day than on other days, which is 15 h. On the other hand, there are 9 fewer hours of solar radiation on December 21.

According to previous studies, the R-value of a 16 cm thick ivy layer is 0.34 m$^2$ k/w [20], and in this article, plant coverage accounts for 50% of the entire façade by the Louver system.

The main façade configuration in the simulation was composed of two layers of 6 mm glass with a 12 mm air layer in the middle filled with argon gas and a metal frame as the structure. The glazed façade specification is depicted in Figure 3.

Apache dynamic simulations in IES VE using the El Prat Barcelona weather file (.epw), which contains data for variables, including dry bulb and wet bulb temperature, wind speed and direction, solar altitude and azimuth, and cloud cover for each hour of the year, were carried out.

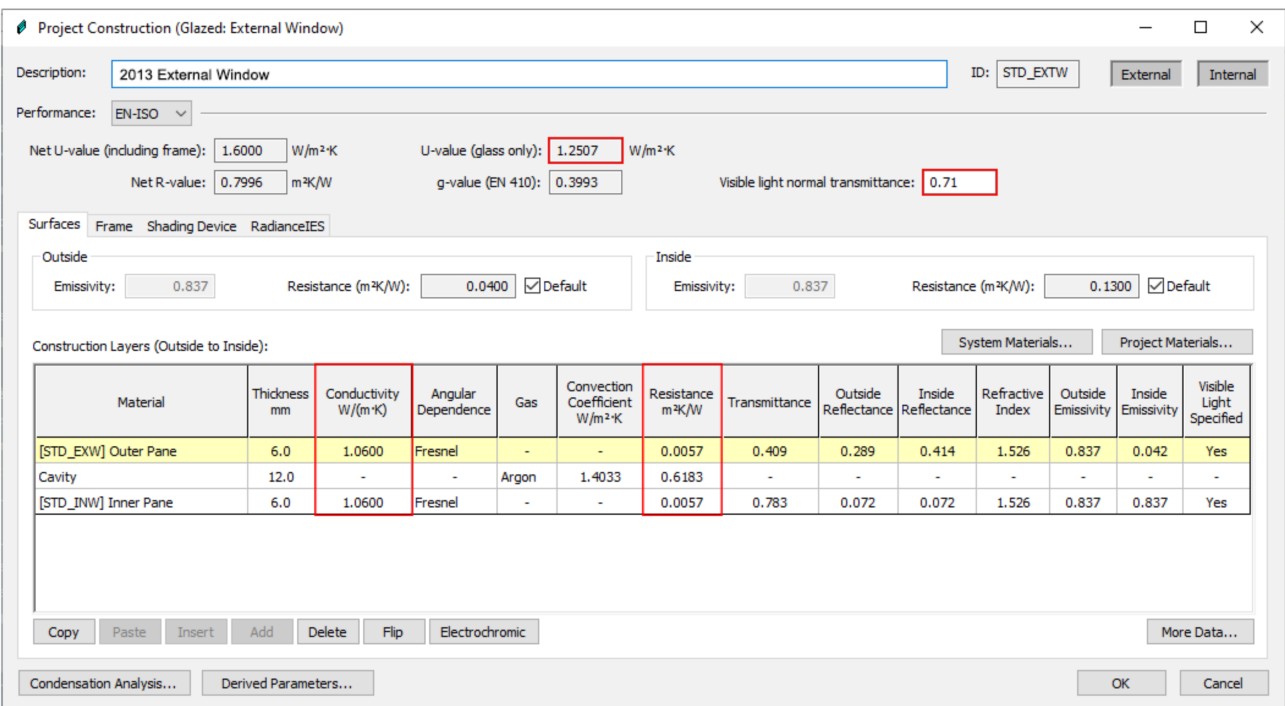

**Figure 3.** Glazed façade specification in IES VE simulation software. © By authors.

### 2.3. Model Characterization

This article's case study model is the Gas Natural Fenosa Building in Barcelona, Spain, which is a high-rise office building. This building has two lower horizontal glazed blocks sticking out and cantilevered from the main tower, which has 22 floors and stands 86 m tall (Figures 4 and 5) and was designed and constructed in 2007 by Enric Miralles and Benedetta Tagliabue.

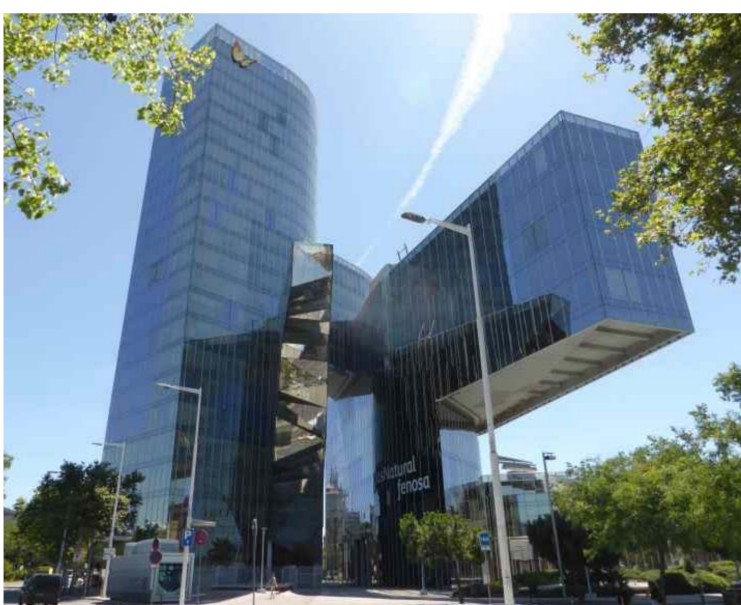

**Figure 4.** West view of Gas Natural Fenosa Building.

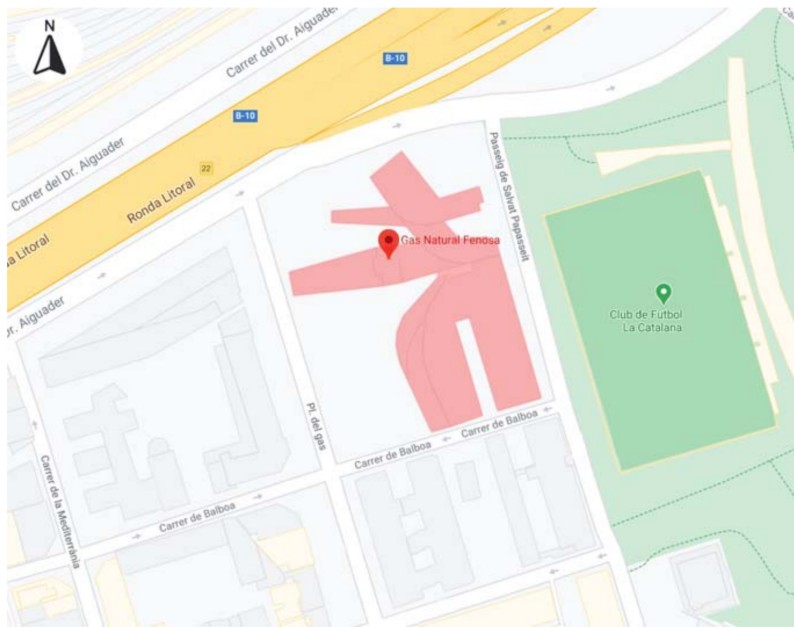

**Figure 5.** The location of the case study. © By authors.

A curtain wall (glazed façade) serves as the continuous façade that connects the three merged buildings. In this article, thermal comfort in both the bare façade (curtain wall) and green façade was analyzed using a building simulation approach in terms of daylight, predicted mean vote (PMV), radiant temperature effects, and air temperature in improving the thermal comfort efficiency of the green façade.

### 2.3.1. Evaluation of Daylight and Solar

The fundamental requirements of building design include providing daylight, reducing glare, and maximizing the thermal efficiency of building façades which, in this study, involved applying a vegetation layer on the building façade as a façade technology. For this purpose, a part of the ground floor (Figure 6) was simulated within IES VE as a function of the RadianceIES (daylighting and electric lighting simulation) package, on a day in summer, which was 21 June at 12 p.m.

Under overcast sky conditions, the daylight factor was calculated as a ratio of internal illuminance on the working plane 85 cm above the floor level to external illuminance on the non-shaded horizontal plane [49,50].

As shown in Figure 6, daylight analysis carried out on a part of the first floor with a southern orientation. Southern façades are more affected by sunlight than other façades, according to a previous study that demonstrated the role of building orientation in green façade performance [18]. As a result of this, the southern façade was chosen for this article. The glazed façade induces solar radiation effects in this open plan office space on sunny summer days, causing the air temperature to increase. Variations in indoor thermal conditions during periods of intense sunlight cause discomfort in the bare façade (glazed).

Barcelona has some of Europe's best winter daylight hours. The average number of daylight hours in December, January, and February is 10 h, while the number of daylight hours in June, July, and August is about 15 h (Table 1). According to Figure 7a–d, which shows the aspect of daylight in terms of thermal conditions in both bare and green façades, comparing each result can help understand the aspect of the green façade that can protect the building from solar radiation during the summer.

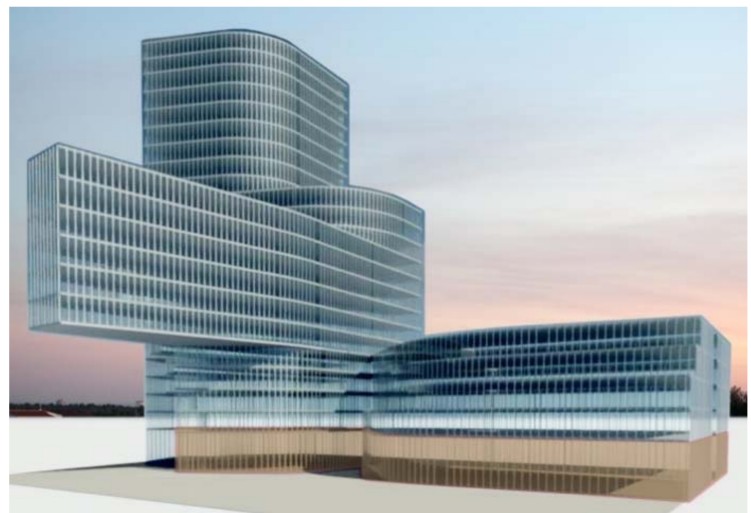

(**a**) 3D view.

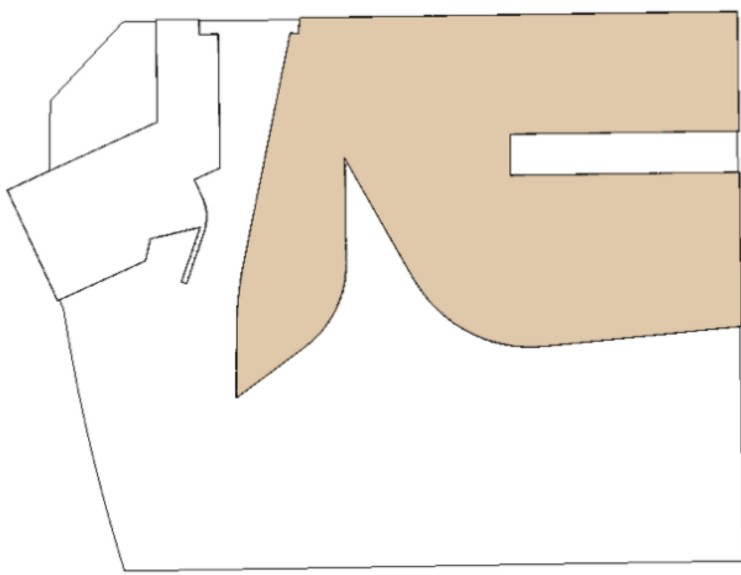

(**b**) Ground floor plan.

**Figure 6.** The portion of the Gas Natural Fenosa Building used for daylight simulation is shown in brown. (**a**) 3D view, (**b**) ground floor plan. © By authors.

**Table 1.** Average hours of daylight in Barcelona.

| Month | Jan | Feb | Mar | Apr | May | Jun | Jul | Aug | Sep | Oct | Nov | Dec |
|---|---|---|---|---|---|---|---|---|---|---|---|---|
| Hours of light | 10 | 11 | 12 | 13 | 15 | 15 | 15 | 14 | 12 | 11 | 10 | 9 |
| Hours of twilight/night | 14 | 13 | 12 | 11 | 9 | 9 | 9 | 10 | 12 | 13 | 14 | 15 |

In analyzing the daylight factor threshold (Figure 7e,f), the value of daylight factor (DF) based on sky component (SC) was 2.00 DF, which, according to CEN European Daylight Standard (EN 17037), is the standard value [51] and, regarding the daylight threshold result:

- Threshold < 2.00 DF = 19.70% in bare façade;
- Threshold < 2.00 DF = 40.94% in green façade.

That is, in the glazed façade, 19.70% of the open space office has less than 2 daylight factors (DFs), indicating that during the day, the workplace has wonderful light and does

not require artificial light, whereas, in the green façade, this value is 40.94%, less than 2 daylight factors (DFs). Thus, using artificial light with a green façade is more advanced than using it with a glazed façade.

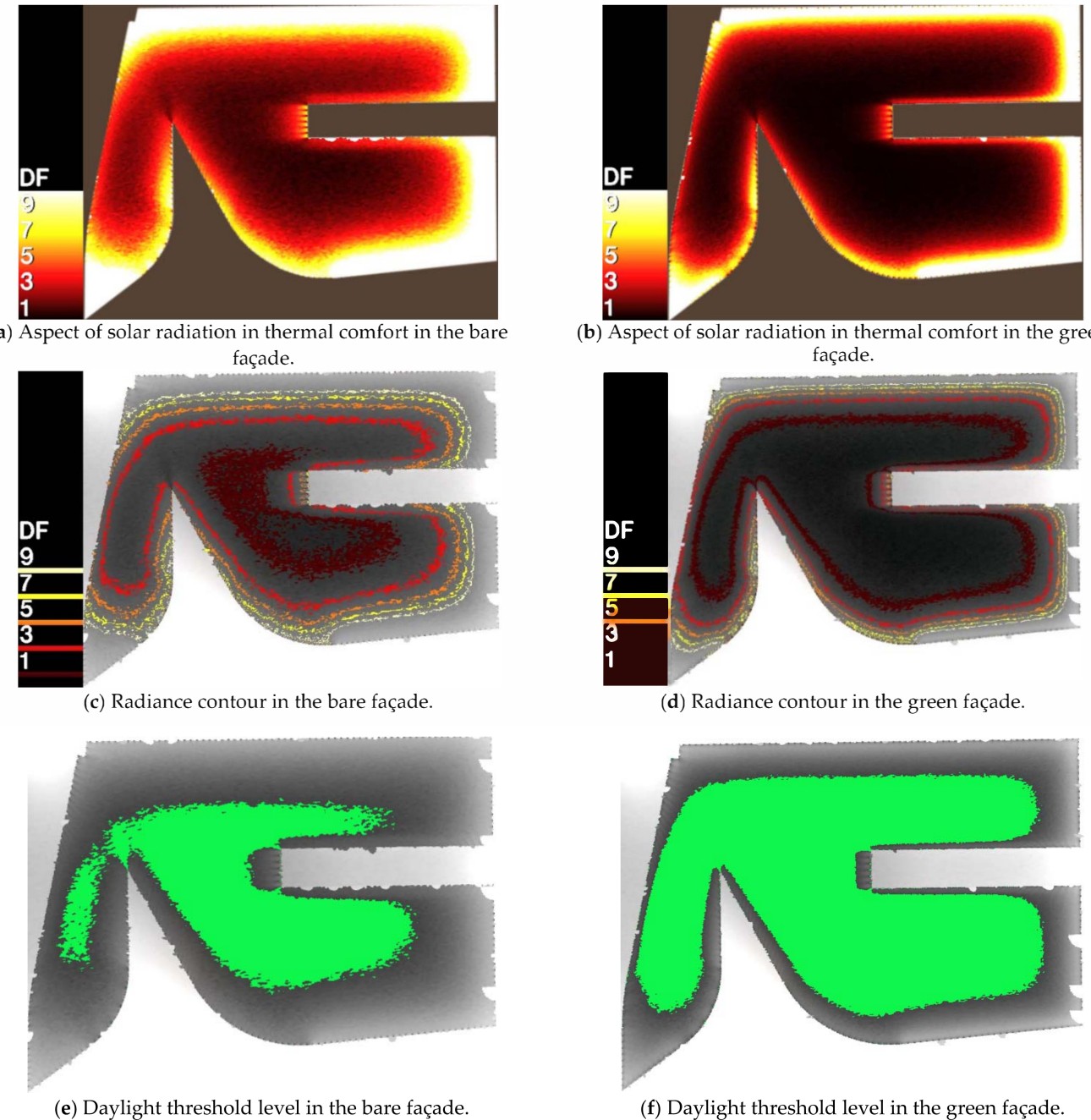

(**a**) Aspect of solar radiation in thermal comfort in the bare façade.

(**b**) Aspect of solar radiation in thermal comfort in the green façade.

(**c**) Radiance contour in the bare façade.

(**d**) Radiance contour in the green façade.

(**e**) Daylight threshold level in the bare façade.

(**f**) Daylight threshold level in the green façade.

**Figure 7.** Daylight analysis of a part of the first floor in terms of daylight and thermal comfort in the bare and green façade on 21 June at 12 p.m. © By authors.

Figures 8 and 9 perfectly illustrate the effect of daylight on thermal comfort with bare (glazed) and green façades. According to the values of the daylight factor in Figures 8 and 9, the vegetation layer will shield the glazed façade from unwanted solar gains during hot summer days by as much as 50%. However, it should be noted that the use of artificial light is needed to protect the glazed façade via the vegetation layer.

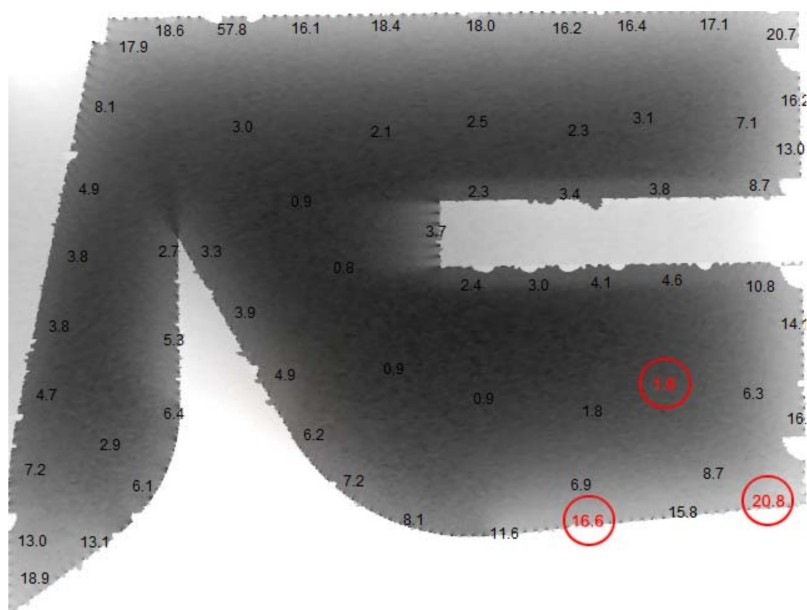

**Figure 8.** The illuminance plan shows the number of daylight factors (SC) on the bare façade. ©
By authors.

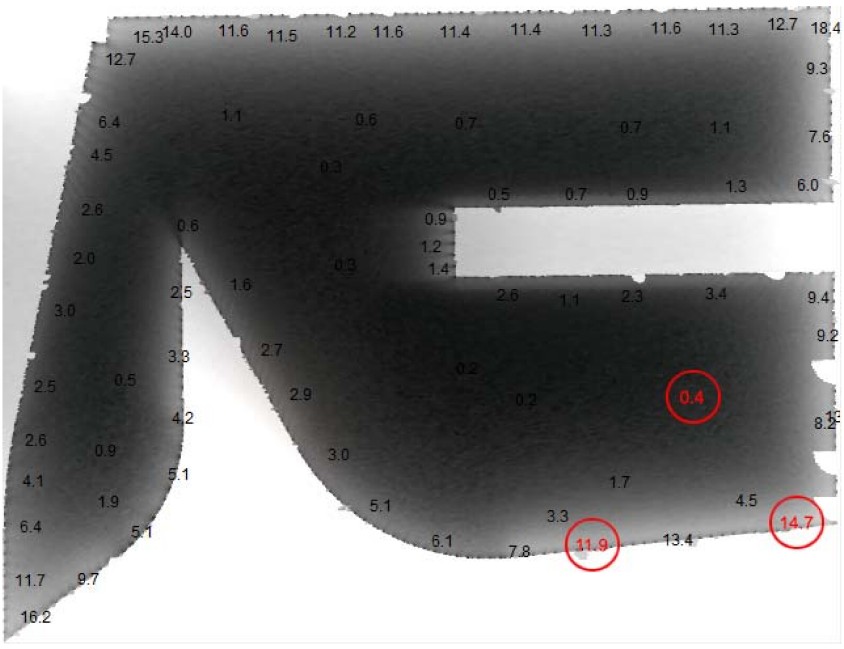

**Figure 9.** The illuminance plan shows the number of daylight factors (SC) on the green façade. ©
By authors.

### 2.3.2. Predicted Mean Vote (PMV)

The predicted mean vote (PMV) is an empirical score of the human sensation of
thermal comfort that has been adopted as an ASHRAE 55 and ISO standard. PMV is a scale
that ranges from −3 to +3, with 0 representing ideal thermal comfort, +3 indicating too hot,
and −3 indicating too cold. In this report, the results were achieved in two days in winter
and summer, 21 December and 21 June.

Another attribute to consider is the predicted percentage of dissatisfaction (PPD), which
is a function of PMV [52]. Figures 10 and 11 show the PMV value of both the bare and green
façades, allowing for a comparison of the bare and green façades in terms of PMV and PPD.
As a result, the PMV value of the green façade is similar to the ideal thermal comfort (0) of the

bare façade (glazing façade). During working hours on 21 December and 21 June, PPD in the green façade is approximately 11.8% and 18%, respectively, but this value was increased by 14.6% and 26% in the bare façade; additionally, the PMV value in the green façade in both figures is closer to the ideal thermal comfort (0) than in the bare façade.

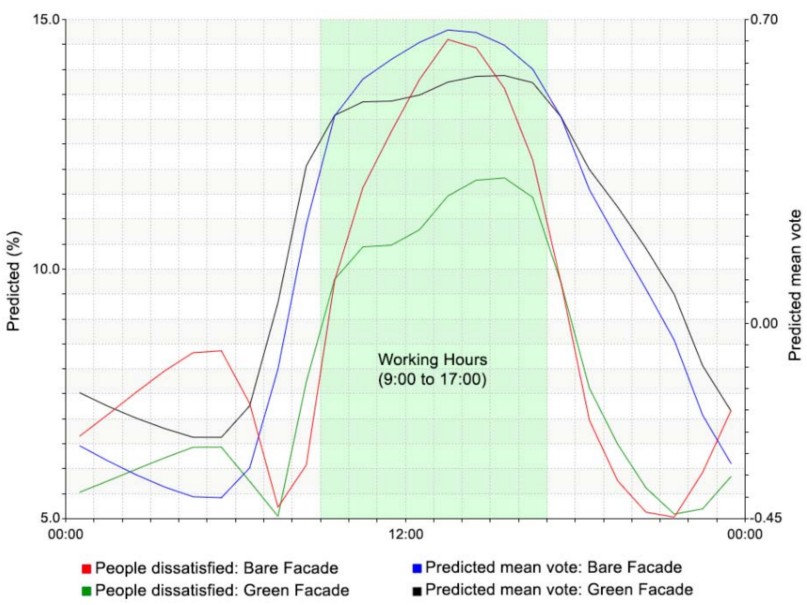

**Figure 10.** Comparison between predicted mean vote (PMV) and the predicted percentage of dissatisfaction (PPD) on 21 December. © By authors.

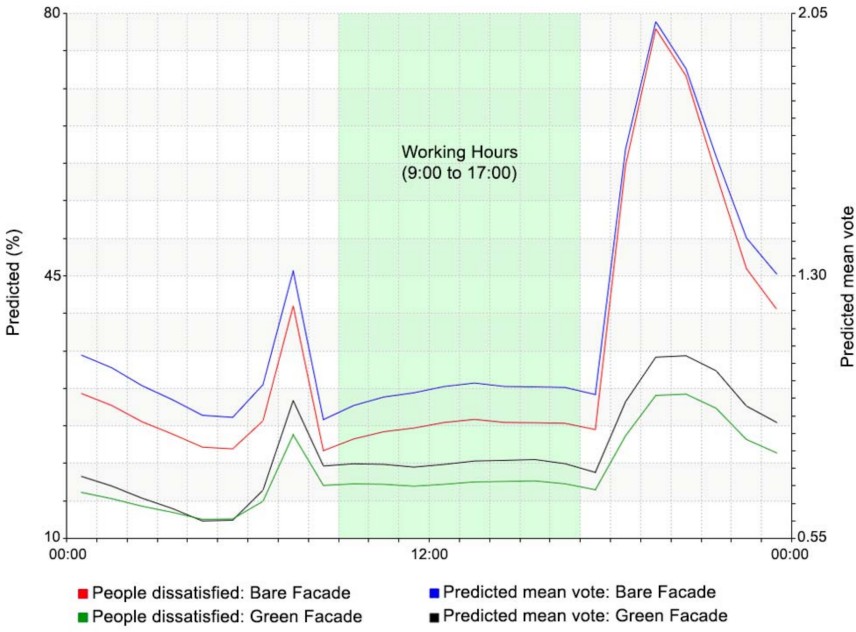

**Figure 11.** Comparison between predicted mean vote (PMV) with the predicted percentage of dissatisfaction (PPD) on 21 June. © By authors.

### 2.3.3. Mean Radiant Temperature

The radiant temperature can be determined using measured temperature values of the surrounding walls and surfaces, as well as their locations in relation to an individual [53]. Spaces with large areas of glass can exhibit greater differences between mean radiant temperature and air temperature. In winter, a cold surface can lead to cool radiant temperatures, and high solar gains can lead to high radiant temperatures since variations

in outdoor temperature and solar radiation during the day affect the radiant temperature and therefore the thermal comfort [54].

Due to the mean radiant temperature on 21 December (Figures 12 and 13), the maximum temperature in both the bare and green façades was 24.5 °C and 23.2 °C, respectively, and on 21 June (Figures 14 and 15), the radiant temperature in the bare façade was 26.6 °C and in the green façade it was 24.3 °C. These variations indicate that a green façade will shield the building envelope from cold weather in the winter and hot weather in the summer, but with different behaviors in each season.

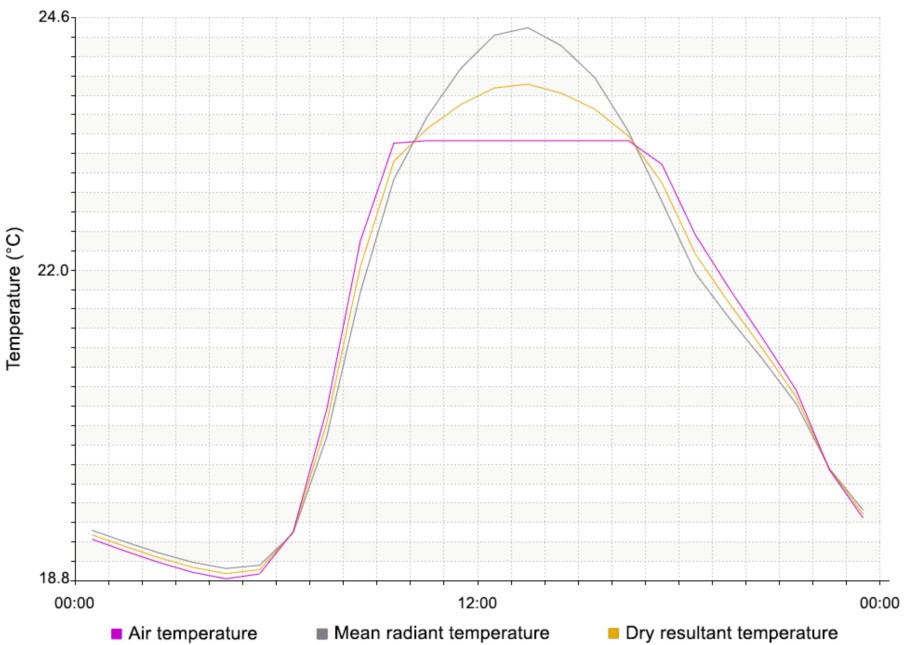

**Figure 12.** Radiant temperature effects on the bare façade on 21 December. © By authors.

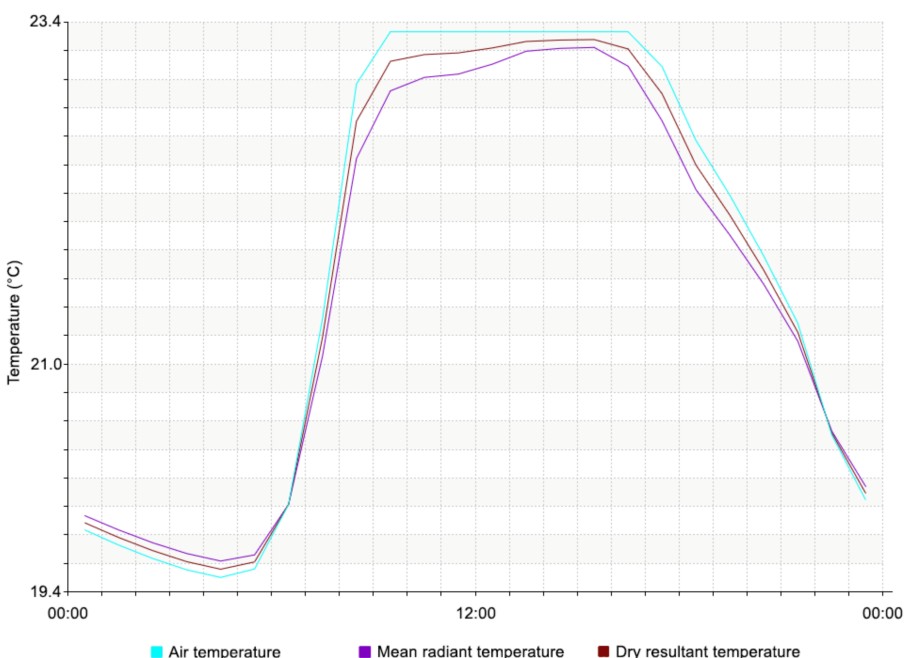

**Figure 13.** Radiant temperature effects on the green façade on 21 December. © By authors.

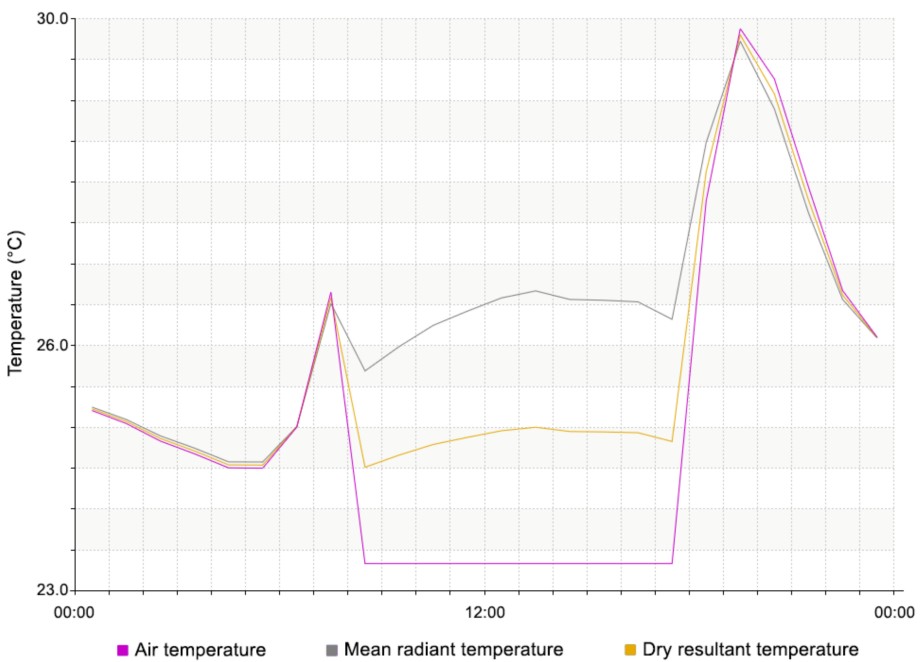

**Figure 14.** Radiant temperature effects on the bare façade on 21 June. © By authors.

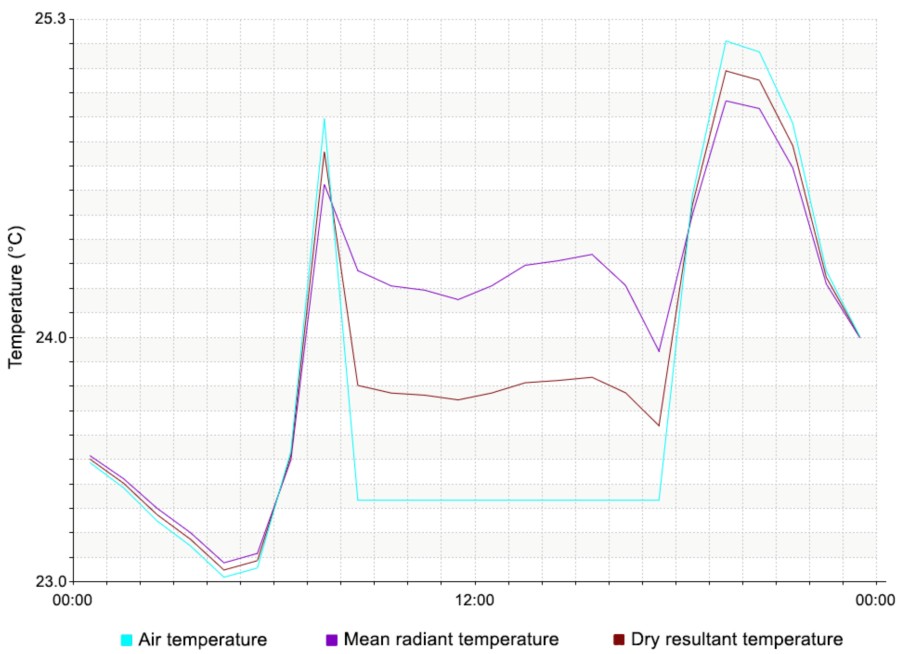

**Figure 15.** Radiant temperature effects on the green façade on 21 June. © By authors.

### 2.3.4. Air Temperature

Figures 16 and 17 show that the difference between the minimum and maximum indoor air temperature during December (winter) in the bare façade is approximately 7 °C; however, this difference is approximately 6 °C in the green façade. In June (summer), this disparity is approximately 9 °C in the bare façade and just 4 °C in the green façade. These values indicate that the indoor air temperature in the green façade is almost the same during the summer (Table 2), meaning that the indoor temperature is optimal and comfortable, allowing air conditioner usage to be reduced, resulting in lower energy use. Furthermore, the difference in air temperature between bare and green façades over a year (Figure 18) demonstrated how much greenery could influence indoor air temperature, thus improving thermal comfort.

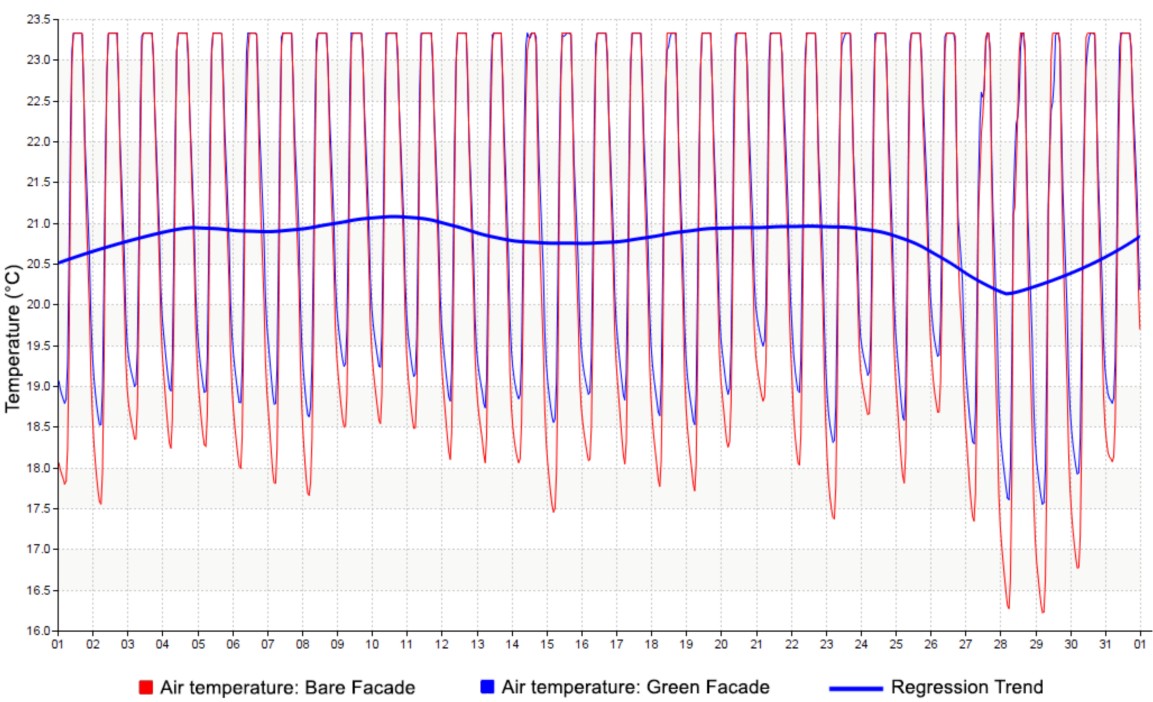

**Figure 16.** Indoor air temperature ranges from 1 December to 31 December in both the bare and green façades. © By authors.

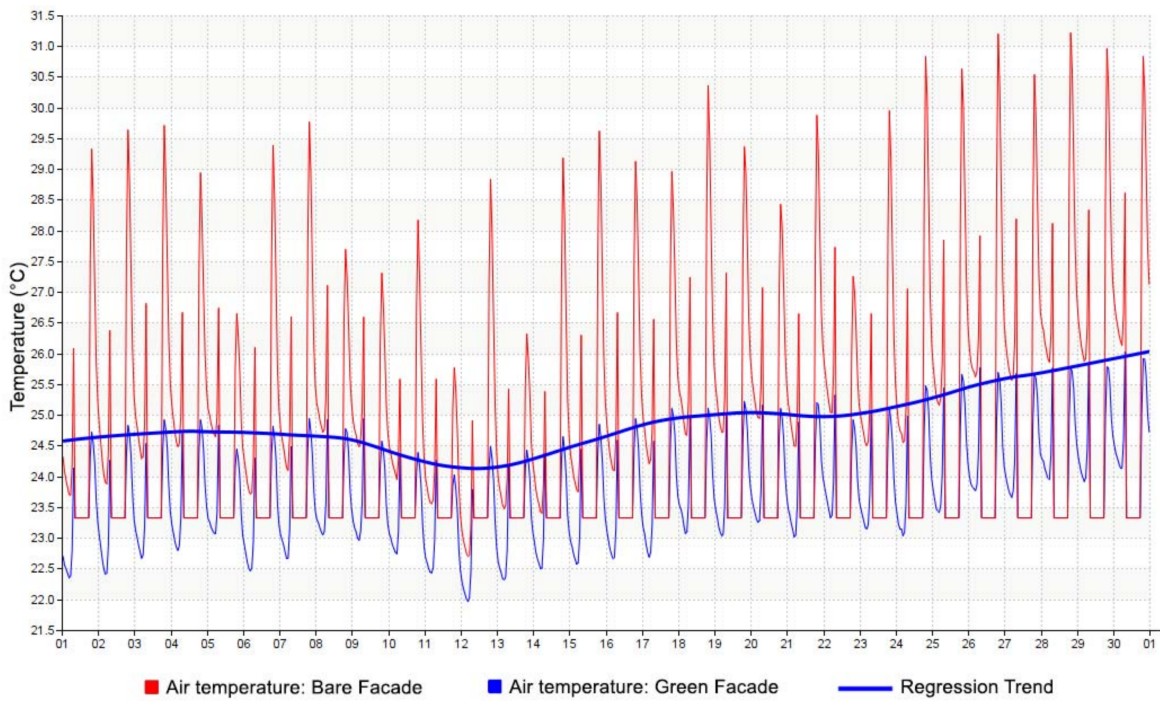

**Figure 17.** Indoor air temperature ranges from 1 June to 31 June in both the bare and green façades. © By authors.

**Table 2.** Indoor air temperature in December and June in both bare and green façades. © By authors.

|  | December | | June | |
|---|---|---|---|---|
|  | **Min** | **Max** | **Min** | **Max** |
| Bare Façade | 16 °C | 23.5 °C | 23.4 °C | 31.3 °C |
| Green Façade | 17.5 °C | 23.5 °C | 22 °C | 26 °C |

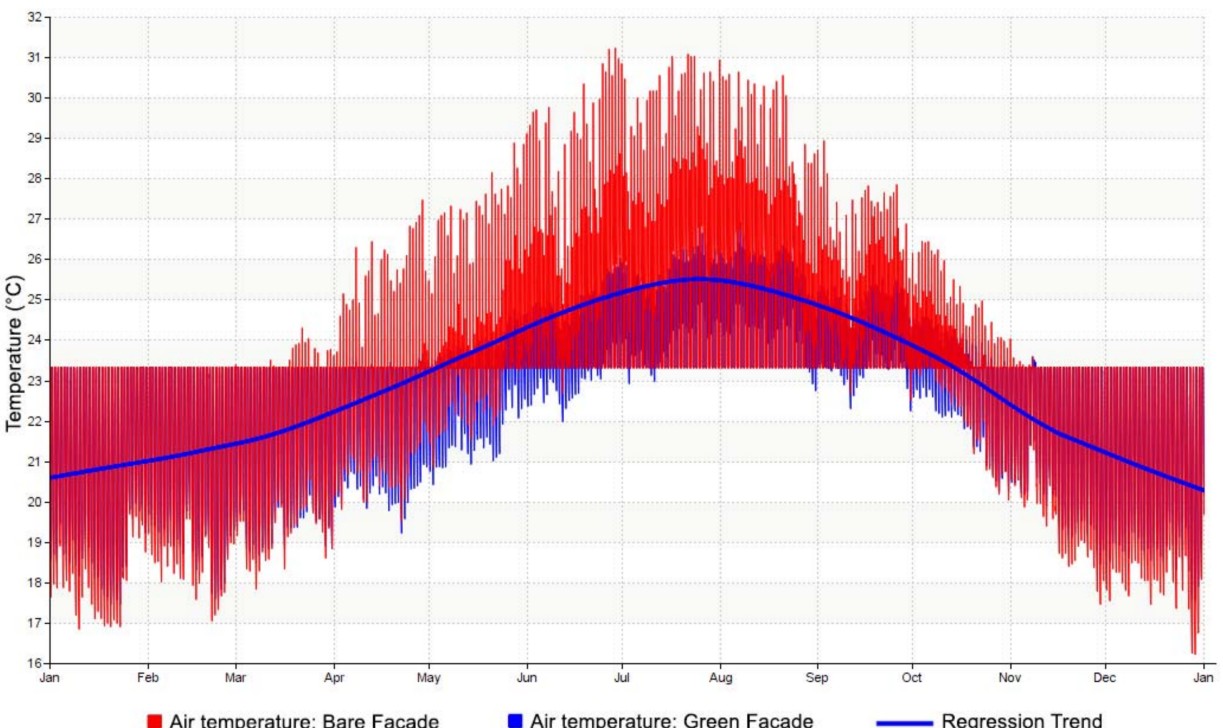

**Figure 18.** Indoor air temperature ranges from 1 January to 31 December in both the bare and green façades. © By authors.

## 3. Results and Discussion

The study analyzed the thermal comfort in the bare and green façades, energy consumption, and various environmental factors for space with the glazing façade on summer and winter days.

By the simulation of the Gas Fenosa Natural Building as a case study, it can be seen that a high radiant temperature in summer in the bare façade (glazed) causes an increase in the PMV. Additionally, it causes indoor thermal discomfort. Due to lowering the radiant and air temperature in the green façade, thermal comfort was improved.

### 3.1. Thermal Comfort

Thermal comfort is described as "that state of mind that expresses satisfaction with the thermal environment" by the international standard EN ISO 7730. In general, it is a comfortable condition in which an individual does not feel overly hot or cold.

By examining the daylight aspect of thermal comfort, predicted mean vote (PMV) and predicted percentage of dissatisfaction (PPD), mean radiant temperature, and air temperature, it is clear that each of these environmental variables could be very successful in improving thermal conditions. These elements are critical in improving living standards and ensuring long-term sustainability on a large scale.

Figures 19 and 20 illustrate the effect of solar radiation on cooling load in the bare (glazed) and green facades. Installing a vertical garden (green façade) on the building's façade could increase thermal comfort. This goal can be accomplished by simply adding a percentage of vegetation to the building's façade; the percentage depends on the building orientation, environmental conditions, building morphology, type of plants, and plant density.

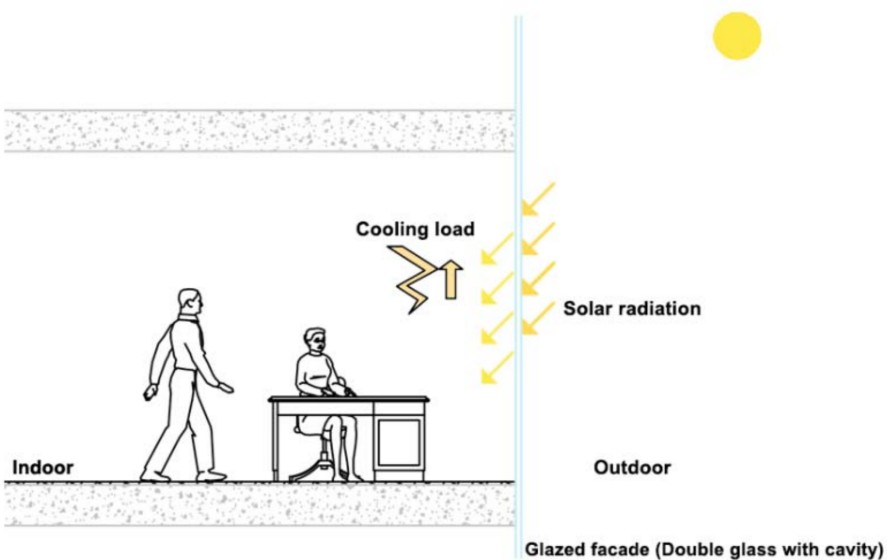

**Figure 19.** Section drawing to better understand the effect of solar radiation on indoor cooling load in the glazed façade. © By authors.

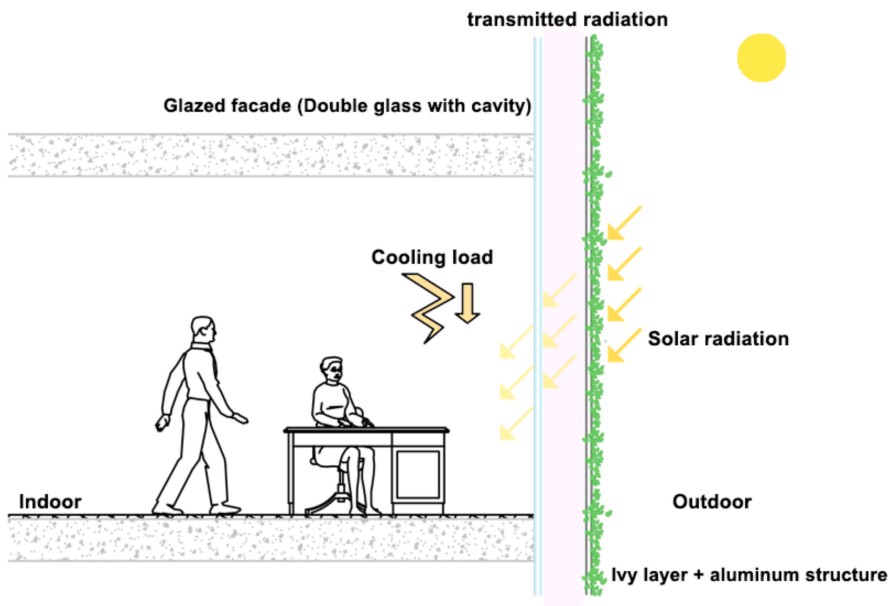

**Figure 20.** Section drawing to better understand the effect of solar radiation on indoor cooling load in the green façade. © By authors.

The increased solar radiation on the façade had a greater impact on the decreased surface temperature of both bare and green façades. As a result, the beneficial effect of the green façade as a second layer, as well as its efficient thermal resistance, increases dramatically with solar radiation; when the solar radiation level is high, the resistance of the plant layer is also high, due to blocked radiation transmission to the exterior wall surface.

As a result, the green façade has many advantages in climates and environments with high levels of solar insolation. The plant layer is also very good at cooling glazed façades that are subjected to high levels of solar radiation.

*3.2. Energy Consumption*

According to the simulation results, by lowering the air temperature in the summer and maintaining the same indoor air temperature in the winter, thermal comfort will be increased, and as a result, energy demand will be substantially reduced (Figure 21).

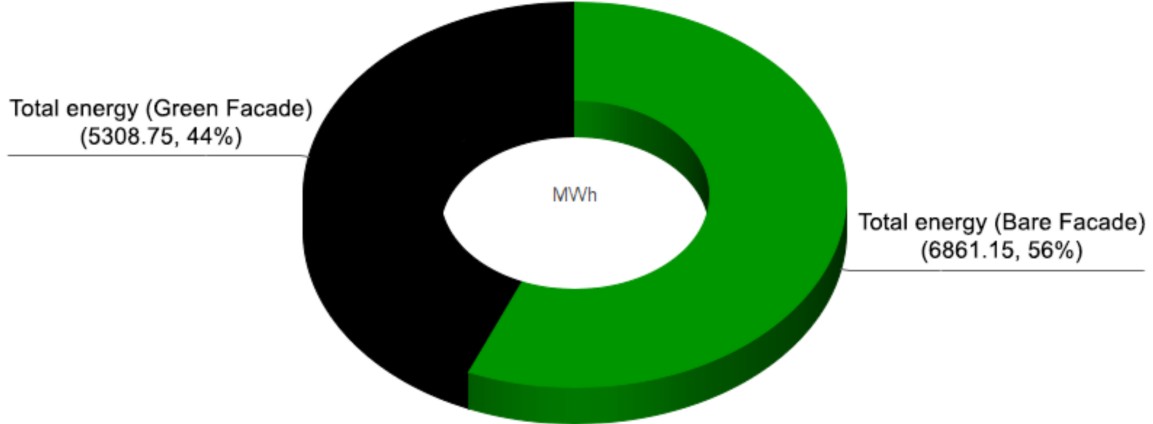

**Figure 21.** Total energy use of the Gas Natural Fenosa Building in one year. © By authors.

Due to high solar radiation during summer, the solar radiation is strong and passes through the glazing façade to indoor spaces with the bare façade, increasing the use of air conditioners and hence energy use.

The building's glazed façade naturally connects the interior with the environment and creates a sense of openness and space and provides light, but when designing a building with a glazed façade, the weather conditions must be considered, and due to Barcelona's weather conditions, the building's façades face about 15 hours of solar radiation during the summer and about 10 hours of solar radiation during the winter. Winter in Barcelona is mild and brief, and the majority of the year is hot. The building's actions will change if a shading layer, such as a green façade, is applied. According to simulation findings, the green façade protected the building façade from extreme solar radiation, resulting in a lower indoor air temperature, radiant temperature, and PMV, as well as lower total energy usage (Figure 21), chiller energy (Figure 22), and total electricity (Figure 23).

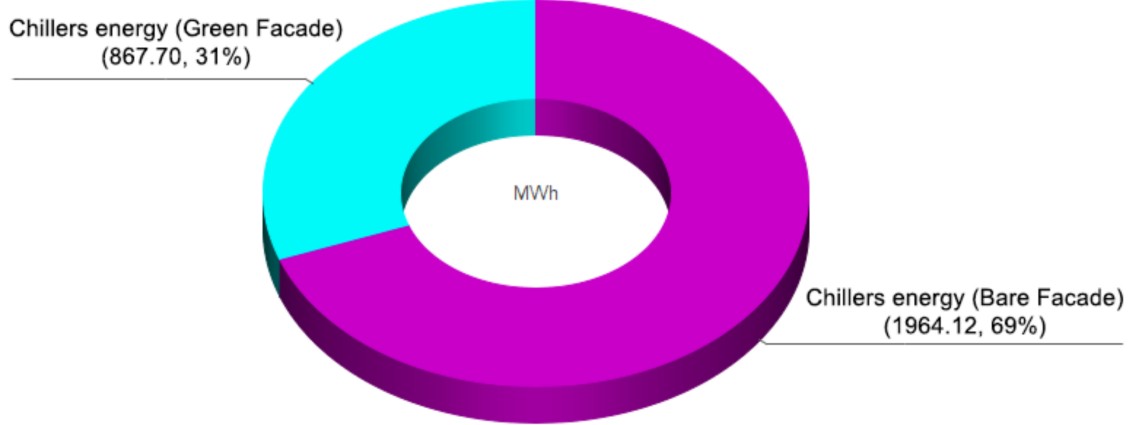

**Figure 22.** Chiller energy use of the Gas Natural Fenosa Building in one year. © By authors.

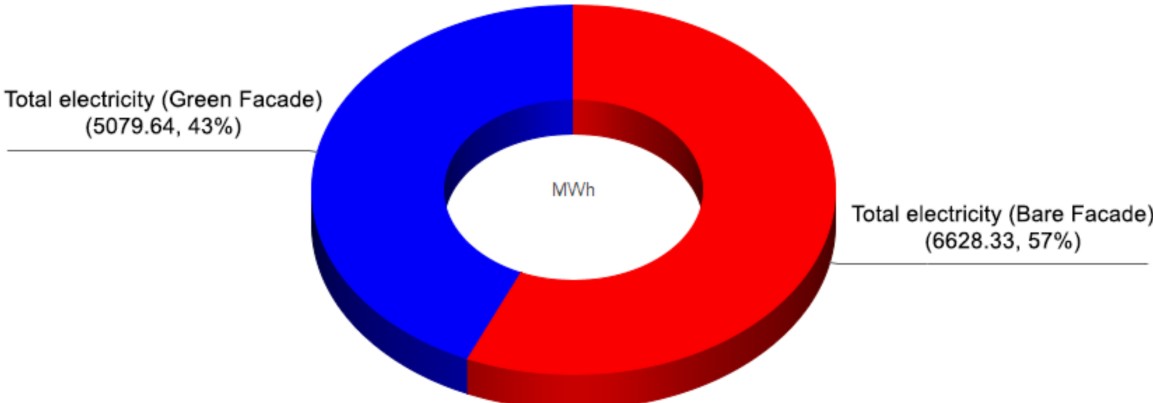

**Figure 23.** Total electricity uses of the Gas Natural Fenosa Building in one year. © By authors.

## 4. Conclusions

Green façades, as a part of urban green infrastructure (UGI) in the sense of vertical greenery, have been described as a collection of human-made elements that provide multiple ecosystem services at the building and urban scales. Regarding the Spanish Regulations for Thermal Facilities in Buildings (RITE), the office building's indoor air quality (IAQ) must be good at a minimum, and this research revealed that by adding a percentage of vegetation on the glazed façade, it is possible to achieve more than good air quality in the workplace while also improving indoor air temperature during the four seasons. Green façades serve many purposes, the most notable of which are improved thermal comfort efficiency, air quality, and building energy savings, as well as the reduction of the urban heat island effect. In this study, IES VE software, as a building simulation tool, was used to simulate the façades of an office building. The simulation model for thermal performance of the green façade as an exterior wall was considered, as well as the evaluation of daylight through the bare façade and vegetation façade, the difference between indoor and outdoor air temperature, mean radiant temperature, the value of predicted mean vote (PMV), predicted percentage of dissatisfaction (PPD), and the investigation of plant R-value. The model was validated using building simulation results from two days, 21 June and 21 December, as well as a one-year analysis that measured thermal comfort efficiency and energy consumption of bare and green façades as a refurbishment option for an existing building located near the coast in Barcelona. The simulation results prove that a vegetation layer on a façade can effectively reduce exterior surface temperatures of façades, daily temperature fluctuations indoors, and overall heat transfer through the exterior wall, particularly on days with high insolation. The results also indicate that green layers with thick leaves (high value of sun resistance) are possibly the most effective in reducing façade surface temperatures and heat transfer through façades, and as a result of this reduction, electricity and overall energy usage in the building can decrease. On hot sunny days, a vegetation layer on the glazed façade was calculated to reduce indoor air temperature by more than 5 °C, providing an efficient R-value that depends primarily on wall orientation, plant type, plant layer density, and green façade structure type.

**Author Contributions:** Conceptualization, F.B.M.; methodology, F.B.M.; software, F.B.M.; validation, J.M.F.M., I.N.D., and E.R.D.; formal analysis, F.B.M.; investigation, F.B.M.; resources, F.B.M.; data curation, F.B.M.; writing—original draft preparation, F.B.M.; writing—review and editing, F.B.M., J.M.F.M., and I.N.D.; visualization, F.B.M.; supervision, J.M.F.M. and I.N.D.; project administration, F.B.M.; funding acquisition, E.R.D. All authors have read and agreed to the published version of the manuscript.

**Funding:** This research was supported by the National Program of Research, Development and Innovation aimed at the Society Challenges with the references BIA2016-77464-C2-1-R and BIA2016-77464-C2-2-R, and two of the National Plans for Scientific Research, Development and Technological Innovation 2013-2016, Government of Spain, titled "Gamificacioñ para la enseñanza del diseño urbano y la integracioñ en ella de la participacioñ ciudadana (ArchGAME4CITY)" and "Diseño Gamificado de visualizacioñ 3D con sistemas de realidad virtual para el estudio de la mejora de competencias motivacionales, sociales y espaciales del usuario (EduGAME4CITY)" (AEI/FEDER, UE).

**Data Availability Statement:** This research did not report any data.

**Conflicts of Interest:** The authors declare no conflict of interest.

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
