# Peer review of "Evaluation of Thermal Comfort Performance of a Vertical Garden on a Glazed Façade and Its Effect on Building and Urban Scale, Case Study: An Office Building in Barcelona"

_sustainability, doi:10.3390/su13126706_

Round 1

Reviewer 1 Report

The authors investigated the impact of adding a vertical greening layer in front of the glazed façade of an office building in Barcelona. The manuscript requires extensive revision; this applies to both content and writing. The scientific quality of the manuscript needs improvement as many important details were either not reported or lacked the required depth. Major scientific issues related to the relationship between lighting and thermal comfort are currently present in the manuscript. The authors seem to mix up radiation with light even though they are two different parameters. Many figures in the manuscript are redundant and could be either removed or merged together. The language of the manuscripts needs substantial improvement. I have pointed out some of the writing issues in the points below but the grammar and writing problems are so many that I could not list all of them here. Moreover, many sentences were unclear and difficult to read. I strongly advise the authors to carefully revise their manuscript. Below are some comments on specific lines in the manuscript.

Line 10: the first and second sentences of the abstract have multiple grammar mistakes.

Line 37: the sentence „with land converted to urban regions projected to triple by 2030 [8]“ is written twice.

Line 41: why is “Environmental Sustainability” capitalized?

Line 45: What do the authors mean by “urban microclimate mitigation”?

Line 48: This paragraph is difficult to read and should be rewritten.

Line 56: This paragraph is also difficult to read and should be rewritten.

Line 59: There are multiple grammar mistakes in this sentence.

Line 62: There is a grammar mistake in this sentence.

Line 63: What do the authors mean by “comprehensive sustainability”?

Line 68: There are multiple grammar mistakes in this sentence.

Line 93: There are multiple writing mistakes in this paragraph.

Line 98: “thermal efficiency, temperature variation and”?

Line 100: Capitalize the first word in the sentence.

Line 109: The results should not be discussed in the Introduction.

Line 111: “Although to mitigate the …” this paragraph is difficult to read.

Line 116: This paragraph belongs to Introduction, not Methods.

Line 138: The authors should provide more details about the requirements of the IDA2 group.

Line 150: Figures 1 and 2 display the …

Line 159: “These simulation results ...” what are the simulations that the word “these” refers to here?

Line 173: What are the properties of the simulated plants (LAD, LAI, etc.)?

Line 177: The authors mention that they validated the model using measured air temperature. However, no information is reported about the results of this validation.

Line 178: It seems that the red rectangles in Figure 3 are incorrectly placed

Line 179: What is meant by the remark “by author”? Do you mean modified, reproduced, etc.? Is the term “by author” generated automatically by the submission system or was this copyright notice added by the authors? If the latter is the case then this should be deleted.

Line 187: Information about the architect and architectural style are irrelevant for the study.

Line 200: What do the authors mean by “daylight thermal”?

Line 212: The authors should refer to Fig. 7, not Fig. 6.

Line 222: The authors should explain what DF/SC means and the source of the standard value of DF = 2.

Line 229: Fig 7 a, b, c, and d are not discussed in the text.

Line 229: Fig 7 c and d: are those temperature counter lines or daylight contour lines? The figure caption says “thermal” and the color scale says “DF”

Line 231: The authors should distinguish between the impact of radiation and the impact of light on thermal comfort. The impact of radiation is physical and related to heat exchange with the body. However, the impact of light is much more complex and is related to physiological responses and subjective thermal perception.

Line 243: Wikipedia is an improper scientific source.

Line 267: The PMV curves reported in Fig 10 and 11 are replicated in Fig 12 and 13, which is redundant.

Line 345: The cooling effect of the plants at night is of course low because there is no solar radiation. This paragraph provides no relevant information.

Line 353: Fig 22 suggests that the plant layer changes the angle of solar radiation, which is of course not correct.

Line 372: It would be interesting to see a comparison of the lighting energy consumption as well.

Line 384: The authors should distinguish between thermal comfort and indoor air quality as they are two separate concepts.

Line 395: There are no investigations of the R-value of the plant in the manuscript. An R-value of 0.34 m2K/W was assumed for the plant layer.

Line 397: Results of model validation are not reported in the manuscript.

Line 412: Parameters such as plant type and density of the plant layer are not investigated in the manuscript and should therefore not be reported in the conclusion. 

Author Response

Dear reviewer,

Thank you for taking the time to read and writing a report for this article. The modifications are as follows:

Line 10: Modified.

Line 37: Modified.

Line 41: Modified.

Line 45: What do the authors mean by “urban microclimate mitigation”? Cities have unintentionally created microclimates such as urban heat islands (UHI), which cause significant problems. Planners and architects are concentrating on climatically responsive urban design to offset or eliminate unfavorable microclimates and foster healthier conditions. This sentence mentioned that the green facade can be used as a solution for decreasing the Urban microclimate.

Line 48: Modified.

Line 56: Modified.

Line 59: Modified.

Line 62: Modified.

Line 63: What do the authors mean by “comprehensive sustainability”? Comprehensive Sustainability is related to Full-Cycle Sustainability: it means that at every stage of a product's (or service's) life, the environmental impact is carefully assessed.

Line 68: Modified.

Line 93: Modified.

Line 98: Modified.

Line 100: Modified.

Line 109: Modified.

Line 111: Modified.

Line 116: This paragraph from methodology moved to the introduction.

Line 138: Added information about IDA and Spanish Regulations for Thermal Facilities in Buildings.

Line 150: Modified.

Line 159: Modified.

Line 173: What are the properties of the simulated plants (LAD, LAI, etc.)? In this case, there was no need to simulate plants, because this research utilized IESVE simulation software, and the R-Value of plants was required.

Line 177: In line 180 there is an explanation about Barcelona's El Prat weather file that Apache dynamic simulations in IESVE using this weather file and measured air temperature automatically.

Line 178: Modified.

Line 179: "By author" means that these figures are created and modified by the author and do not copy from other articles or books.

Line 187: Modified.

Line 200: The meaning of Daylight Thermal is the amount of daylight that has an effect on people's thermal perception, resulting in a cross-modal effect. The sentence has been modified.

Line 212: The sentence refers to Figure 6, and that’s right.

Line 222: Modified.

Line 229: Modified (line 226).

Line 229: According to IES VE software, this is the Radiance contour image, and the caption modified.

Line 231: Actually, visible 'light' is a form of radiation that can be described in electromagnetic waves as an energy moving. I modified 2.2.1. Evaluation of the Daylight section of the article.

Line 243: Modified.

Line 267: Figure 10 and 11 deleted.

Line 345: That line deleted.

Line 353: Modified.

Line 372: The usage of total light is not related to this article.

Line 384: Modified.

Line 395: Plant simulation has not been within the scope of this research and because of that we are referred to validated research by Irina Susorova, Melissa Angulo, Payam Bahrami, and Brent Stephens (Ref. 20).

Line 412: In the model validation section of article lines 176-178, those parameters have been described. “The plants on the facade covered 50% of the whole facades by the Louver structure and considering 16 cm layer of the plant (ivy) and according to previous research, the R-Value of 16cm thick ivy layer is 0.34 m2k/w.”

The English language of this article has also been modified.

Best regards,

Faezeh

Reviewer 2 Report

This is a good paper. Please try to improve the quality of table and figures because some of them are blur or distorted. The language can be improved, so please do a careful proofreading.  

Author Response

Dear reviewer,

Thank you for taking the time to read and writing a report for this article. The article has been modified.

The English language of this article has also been corrected.

Best regards,

Faezeh

Reviewer 3 Report

The goal of the paper was  to study comfort performance in the case of building with both glazed and green façade and its assessment. For that reason it has been simulated a case study being a building of Barcelona, whereas IESVE has been used as the simulation tool.

The studies were conducted properly. However, the obtained results could be compared with the results of similar studies.

The last sentence of the abstract requires some editing.

Author Response

(The authors gave the same response as above.)

Round 2

Reviewer 1 Report

The authors have significantly improved the manuscript and responded well to most of my previous comments. However, some of my comments were answered without including the answer in the manuscript e.g. the meaning of comprehensive sustainability. Furthermore, the authors replied to my comments without reporting my questions which meant I had to go back and forth between my comments file and their replies file. I strongly urge the authors not to do this in the future. Some minor spelling mistakes can still be found here and there, but the language of the text is now clear and readable. I recommend accepting the article with few more revisions:

Line 44: The concept of urban microclimate mitigation should be defined in the manuscript.

Line 60: The concept of comprehensive sustainability should be defined in the manuscript.

Line 123: The greening increases the R-value but reduces the U-value.

Line 160: Regarding my previous comment “The authors mention that they validated the model using measured air temperature. However, no information is reported about the results of this validation”: the authors should present a diagram comparing the measured and simulated results directly.

Author Response

Dear reviewer,

Thank you for taking the time to read my article again. I revised the manuscript in response to your feedback.

Line 44: The concept of urban microclimate mitigation should be defined in the manuscript. Answer: Modified. Lines 44-47 of the article have provided a description of the microclimate and urban microclimate.

Line 60: The concept of comprehensive sustainability should be defined in the manuscript. Answer: Modified. The idea of comprehensive sustainability is defined in lines 64-69 of the article.

Line 123: The greening increases the R-value but reduces the U-value. Answer: The sentence has been modified.

Line 160: Regarding my previous comment "The authors mention that they validated the model using measured air temperature. However, no information is reported about the results of this validation ": the authors should present a diagram comparing the measured and simulated results directly. Answer: In section 2.2.3. Air Temperature, the air temperature of the bare façade, which serves as the main façade, and the green façade, which acts as a secondary layer on the façade, have been simulated, measured, and compared. The validation results are shown in Figures 16, 17, and Table 2.

The English language has been improved.

Kindest regards,

Faezeh Bagheri
